# Intraperitoneal Injection of MCC950 Inhibits the Progression of Myopia in Form-Deprivation Myopic Mice

**DOI:** 10.3390/ijms242115839

**Published:** 2023-10-31

**Authors:** Zhengyu Chen, Kang Xiao, Qin Long

**Affiliations:** 1Department of Ophthalmology, Peking Union Medical College Hospital, Chinese Academy of Medical Sciences & Peking Union Medical College, Beijing 100730, China; chenxunan1@outlook.com (Z.C.); b2021001172@pumch.cn (K.X.); 2Key Laboratory of Ocular Fundus Diseases, Chinese Academy of Medical Sciences, Beijing 100730, China

**Keywords:** NLRP-3, myopia, MCC950

## Abstract

Myopia, one of the most prevalent ocular diseases worldwide, is projected to affect nearly half of the global population by 2050. The main cause of myopia in most patients is axial myopia, which primarily occurs due to the elongation of the eyeball, driven by changes in the extracellular matrix (ECM) of scleral cells. Previous studies have shown that NLRP3, an important inflammatory mediator, plays a critical role in regulating the expression of MMP-2 in the sclera. This, in turn, leads to a decrease in the expression of Collagen-1, a major component of the scleral ECM, triggering the remodeling of the scleral ECM. This study aimed to investigate the effect of MCC950, an inhibitor of NLRP3, on the progression of myopia using a mouse form-deprivation myopia (FDM) model. The FDM mouse model was constructed by subjecting three-week-old C57BL/6J mice to form-deprivation. The mice were divided into experimental (*n* = 10/group; FDM2M, FDM4M, FDM2W, and FDM4W) and control groups (*n* = 5/group). The experimental groups were further categorized based on the duration of form deprivation (2 and 4 weeks, labeled as 2 and 4, respectively) and the type of injection received (MCC950 or saline, labeled as M and W, respectively). MCC950 was injected at a concentration of 50 mg/mL, with a dose of 10 mg per kilogram of body weight. Meanwhile, the saline group received the same volume of saline. Refraction and axial length measurements were performed for each eye. The expression levels of NLRP3, caspase-1, IL-1β, IL-18, MMP-2, and Collagen-1 in the sclera were assessed using immunohistochemistry and Western blotting. The intraperitoneal injection of MCC950 did not significantly affect refraction or axial length in normal mice (*p* > 0.05). However, in FDM mice, MCC950 attenuated the elongation of the axial length and resulted in a smaller shift towards myopia compared to the saline group (FDM4M vs. FDM4W, *p* = 0.03 and *p* < 0.05, respectively). MCC950 decreased MMP-2 expression (*p* < 0.05) but increased Collagen-1 expression (*p* < 0.05) in the experimental eyes when compared to the saline group. Within the MCC950 group, the expression of MMP-2 was increased in the experimental eyes at 4 weeks (*p* < 0.05), while that of Collagen-1 was decreased (*p* < 0.05), which is consistent with changes in refractive error. Immunohistochemical analysis yielded similar results (*p* < 0.05). MCC950 also reduced the expression levels of NLRP3 (*p* = 0.03), caspase-1 (*p* < 0.05), IL-1β (*p* < 0.05), and IL-18 (*p* < 0.05) in the experimental eyes compared to the saline group. Within the MCC950 group, the expression levels of NLRP3 and caspase-1 were comparable between the experimental and control eyes (*p* > 0.05), whereas IL-18 expression was higher in experimental eyes (*p* < 0.05). IL-1β expression was higher in the experimental eyes only at week 4 (*p* < 0.05). The intraperitoneal injection of MCC950 can inhibit the progression of myopia in FDM mice, possibly by regulating collagen remodeling in the sclera through the NLRP3-MMP-2 signaling pathway. Therefore, MCC950 holds promise as a potential therapeutic agent for controlling the progression of myopia.

## 1. Introduction

Myopia is a prevalent eye condition globally and a significant cause of vision loss [1]. The most common type of myopia is axial myopia, which is characterized by the continuous remodeling of the sclera’s extracellular matrix (ECM), leading to decreased tissue hardness and increased elasticity [2,3]. The sclera primarily consists of collagen fibers, with Collagen-1 being the predominant type, accounting for approximately 50–70% of the sclera’s weight, along with Collagen-3 and Collagen-4 [4,5]. In myopic eyes, there is a downward trend in the expression of Collagen-1 in the scleral ECM. Previous studies have demonstrated that matrix metalloproteinases (MMPs) contribute to extracellular matrix (ECM) remodeling by degrading Collagen-1 in the sclera [6,7]. Ge et al. conducted research showing that MMP-2 deficiency reduced myopia progression by 59% compared to the control group in a form-deprivation myopia (FDM) model [8]. Moreover, interleukin-18 (IL-18) and interleukin-1β (IL-1β) have been found to promote the upregulation of MMP-2 expression in various cells, and inflammatory responses can elevate the expression of these interleukins [9,10]. The research conducted by Huang et al. in the field of proteomics suggests that the complement system can influence the development of myopia by affecting inflammatory responses [11]. NOD-, LRR-, and Pyrin domain-containing protein 3 (NLRP3) is an intracellular sensor capable of detecting various microbial patterns, endogenous danger signals, and environmental stimuli. It can self-assemble to form an inflammasome, triggering an inflammatory response under different external stimuli. Previous research has shown that NLRP3 can enhance the expression of Caspase-1, leading to the release of IL-18 and IL-1β [12,13]. Therefore, there may exist a signaling pathway in the sclera involving NLRP3↑ → Caspase-1↑ → IL-18↑ or IL-1β↑ → MMP-2↑ → Collagen-1↓, which contributes to the progression of myopia. In our previous study, we confirmed that changes in the expression levels of NLRP3 in the FDM mouse model do indeed affect the progression of myopia [14]. These findings support the notion that the NLRP3 pathway plays a role in myopia development and highlight its potential as a target for intervention in myopia management.

MCC950 is a compound derived from diacylsulfonamide and is a highly specific inhibitor of NLRP3 [15]. It has been extensively studied for its ability to modulate the NLRP3 inflammasome, a key component of the innate immune system. Coll et al. discovered that the primary mechanism of action for MCC950 is to inhibit the activation steps of the NLRP3 inflammasome. Importantly, it does not interfere with essential cellular processes such as K^+^ efflux, Ca^2+^ influx, or the interaction between NLRP3 and ASC. Furthermore, it does not exhibit significant inhibitory effects on the activation of other inflammasomes, including Absent in melanoma 2 (AIM2), NLR family CARD domain-containing protein 4 (NLRC4), or nucleotide-binding oligomerization domain-like receptor protein 1 (NLRP1) [16]. The high specificity of MCC950 has made it a valuable tool in studying diseases associated with NLRP3 activation. For instance, Zhang et al., demonstrated that MCC950 can protect retinal endothelial cells from functional impairment under high-glucose conditions by downregulating the NLRP3 pathway [17]. Yujian et al. found that MCC950 can reduce the apoptosis of ganglion cells under chronic high-intraocular-pressure conditions [18]. Initially, MCC950 was discovered to inhibit the activation of the NLRP3 signaling pathway in bone marrow-derived macrophages in humans and mice [15]. Subsequent studies, however, revealed that it can also antagonize NLRP3 expression in various cell types, including microglia, muscle cells, and dendritic cells [19,20,21]. Correspondingly, studies utilizing MCC950 as a therapeutic intervention have emerged in these specific cell types.

Considering the role of NLRP3 activation and inflammation in the progression of myopia, it is reasonable to hypothesize that MCC950 could potentially reduce the inflammatory response associated with myopia by inhibiting the expression of NLRP3, thereby slowing down disease progression. To verify this hypothesis, our current study aimed to assess the impact of MCC950 on the progression of myopia using a form-deprivation myopia (FDM) mouse model. Additionally, we assessed the expression levels of NLRP3 pathway-related cytokines to gain insights into the underlying mechanism. Considering the lack of effective drugs for treating myopia and the growing number of myopia patients, this study aims to offer preliminary experimental evidence for utilizing NLRP3 inhibitors to regulate myopia progression. Our goal is to establish a foundation for potential future applications of these inhibitors in the treatment of human myopia in the long run.

## 2. Results

### 2.1. Intraperitoneal Injection of MCC950 Does Not Significantly Influence the Refractive Status of Control C57BL/6j Mice

To assess the effects of MCC950 on the refractive status of mice, we first investigated its impact on normal mice without the form-deprivation model. We compared the refractive status and expression levels of relevant factors in three groups: mice receiving alternate-day intraperitoneal injections of MCC950, mice receiving alternate-day intraperitoneal injections of saline, and a control group that did not receive any intraperitoneal injections. Among the age-matched groups, no statistically significant differences were observed in refractive status or axial length among the three groups (Figure 1A,B). Immunohistochemical analysis revealed that the expression level of NLRP3 was significantly lower in the FDM4M group compared to the FDM4W group (*p* = 0.03). However, the differences in NLRP3 expression between the non-injected group and the MCC950-injected group, as well as between the non-injected group and the saline-injected group, did not reach statistical significance. Similar patterns were observed for other factors associated with the NLRP3 signaling pathway, such as IL-1β, IL-18, and caspase-1, but none of these differences were statistically significant (Table 1). Therefore, we can conclude that, in the absence of FPM construction, intraperitoneal injections of MCC950 or saline in wild-type C57BL/6j mice of the same age do not significantly affect the refractive status. It is worth noting that alternate-day intraperitoneal injections may lead to the upregulation of NLRP3 and its pathway factors, but the extent of this upregulation does not reach statistical significance.

The values in the table represent the average immunoreactivity scores of five eyes within the same group, and the data are expressed as mean ± standard deviation. The designation “2x” indicates that the eyes in this group of mice are from the blank control group of the same age as the FDM2M group. Similarly, “2y” indicates that the eyes are from the blank control group of the same age as the FDM2W group, and “2z” indicates that the eyes are from the blank control group of the same age as the FDM2 group without intraperitoneal injection, and so on.

### 2.2. MCC950 Attenuates the Myopia-Promoting Effect of FDM Mice

After clarifying the impact of intraperitoneal injection on mouse refractive status, we proceeded to investigate the effect of MCC950 on the refractive status of form-deprived mice. We compared the refractive power, axial length, and expression levels of MMP-2 and Collagen-1 in the experimental eyes of the MCC950 group, the control eyes of the MCC950 group, and the experimental eyes of the saline group, all of the same age.

In terms of refractive power, a trend emerged where the refractive power of the saline group’s experimental eyes was lower than that of the MCC950 group’s experimental eyes, and the refractive power of the MCC950 group’s control eyes was in between. In the FDM2 group, there was no statistical difference among the three groups, whereas in the FDM4 group, a statistically significant difference was observed (Figure 2). Conversely, the axial length showed an opposite pattern, with the saline group’s experimental eyes having a greater axial length compared to the MCC950 group’s experimental eyes and the MCC950 group’s control eyes. However, the only statistically significant difference was observed between the experimental eyes of the FDM4M group and the FDM4W group (*p* = 0.03).

By analyzing the expression levels of cytokines using Western blotting (Figure 3), we observed that in mice of the same age, the expression level of MMP-2 was significantly higher in the experimental eyes of the saline group compared to the MCC950 group, while the expression of Collagen-1 showed an opposite trend. Within the MCC950 group of the same age, the control eyes consistently exhibited higher Collagen-1 expression compared to the experimental eyes, and while the difference in MMP-2 expression between the two eyes was not significant at two weeks; it significantly increased in the experimental eyes at four weeks, reflecting the myopia progression in each eye.

In addition, we corroborated these findings through immunohistochemistry. The images visually confirmed results similar to those observed in the Western blot analysis (Figure 2). Furthermore, the conversion of these images into semi-quantitative immunoreactivity scores closely mirrored the results obtained from the Western blot analysis (Table 2). The main difference was observed in the two-week-old group, where immunohistochemistry showed comparable Collagen-1 expression between the control eyes and experimental eyes of the MCC950 group, both significantly higher than the experimental eyes of the saline group. In Western blotting, however, the expression of Collagen-1 in the control eyes of the MCC950 group was significantly higher than in the other two groups.

Collectively, the results indicate that in FDM mice, the experimental eyes exhibited a clear myopia shift compared to the control eyes, with a more pronounced effect observed with longer duration of form-deprivation. Compared to the experimental eyes of mice injected intraperitoneally with saline, those injected with MCC950 showed a reduced degree of myopia shift, lower expression levels of MMP-2, and higher expression of Collagen-1. These findings suggest that MCC950 has an inhibitory effect on the myopia-promoting effect of the FDM model.

The values in the table represent the average immunoreactivity scores of five eyes within the same group. Data are expressed as mean ± SD. The label “2c” corresponds to the control eyes in the FDM2M group, “2e” represents the experimental eyes in the FDM2M group, “2b” denotes the experimental eyes in the FDM2W group, and so on for the remaining groups.

### 2.3. MCC950 Inhibits the Expression Levels of NLRP3 and Its Pathway-Related Factors

After confirming the effect of MCC950 on the refractive, axial and myopia-related factor in FDM mice, we proceeded to investigate its impact on the expression of factors associated with the NLRP3 signaling pathway.

First, we compared the expression levels within the MCC950 group. In mice of the same age, the expression levels of NLRP3 were similar between the experimental eyes and control eyes. The experimental eyes showed slightly higher expression of Caspase-1 compared to the control eyes, although this difference was not statistically significant. The expression level of IL-18 was significantly higher in the experimental eyes compared to the control eyes in both age groups, while the difference in IL-1β expression was only observed in the four-week-old group (Figure 3).

Next, we compared the differences between the experimental eyes of the MCC950 group and the saline group. The results showed that, except for a similar difference in IL-1β expression between the two groups of experimental eyes in the two-week-old group, all other factors (NLRP3, Caspase-1, IL-18, and IL-1β) in the same age group exhibited significantly higher expression levels in the experimental eyes of the saline group compared to the MCC950 group. The immunohistochemistry results also supported this observation (Figure 2). Therefore, it is evident that form deprivation leads to an increase in the expression levels of various factors in the NLRP3 signaling pathway, while MCC950 exerts an inhibitory effect on this increase.

## 3. Discussion

In our previous study [14], we demonstrated the involvement of NLRP3 signaling pathway-related cytokines in the progression of myopia in FDM modeling using both wild-type mice and NLRP3−/− mice. In the present study, we aimed to further explore whether the NLRP3 inhibitor MCC950 could attenuate the progression of myopia in mice.

Since this study involved intraperitoneal drug injection, we first examined the impact of simple intraperitoneal injection on the expression levels of NLRP3 signaling pathway factors without FDM modeling. The experimental results revealed that frequent intraperitoneal injection behavior (once every other day) itself caused a mild inflammatory response in the sclera. The expression levels of NLRP3 and related factors in the saline injection group of the same age were higher than those in the blank control group without injection, although the difference was not statistically significant. However, this mild inflammation did not have a significant impact on the refractive development of mice without FDM modeling.

Under the premise that the injection behavior itself triggered an inflammatory response, the expression levels of NLRP3, Caspase-1, IL-1β, and MMP-2 in the MCC950 injection group were still lower than those in the blank control group and significantly lower than those in the saline injection group. This indicates that MCC950 can exert an inhibitory effect on the NLRP3 signaling pathway in the mouse sclera. Subsequently, we compared the refractive status and the expression levels of FDM mice in the saline injection group and MCC950 injection group at different ages. We observed that the degree of myopic drift in the experimental eyes of the saline injection group was higher than that in the MCC950 injection group in each age group. In addition, NLRP3, Caspase-1, IL-18, and IL-1β also exhibited similar trends. These findings not only confirmed the conclusions obtained in our previous study, which indicated that the high expression of NLRP3 signaling pathway-related factors affects the progression of mouse myopia, but also demonstrated that in FDM mice, MCC950 could influence the progression of myopia by inhibiting the expression of the NLRP3 signaling pathway.

The mice in this study were initiated at the age of 3 weeks ± 3 days, which roughly corresponds to 8 years of age in humans [22]. This stage is considered crucial for the onset and development of myopia. Currently, several methods are employed in clinical practice to slow down the rapid progression of myopia in children, including atropine, pirenzepine, orthokeratology, defocus lenses, rigid gas-permeable contact lenses, invisible glasses, and progressive lenses. A meta-analysis has shown that drug interventions, specifically muscarinic antagonists like atropine and pirenzepine, are the most effective options [23]. However, our understanding of the mechanisms underlying the occurrence and development of myopia remains incomplete, and effective therapeutic strategies targeting the etiology of myopia are still lacking. MCC950 has been extensively studied for its ability to inhibit NLRP3 expression and thereby control disease progression. In this study, we investigated its inhibitory effect on myopia progression by regulating the NLRP3 pathway in the sclera. If this effect can be further confirmed in future studies, MCC950 may become a potential option for controlling myopia progression in the future.

The limitations of this study primarily lie in the fact that the FDM-modeling process requires the eyes to be covered, making it difficult to administer MCC950 as eye drops or periocular injections. Additionally, the final experimental results are affected by systemic errors due to changes in the expression levels of inflammatory factors caused by frequent intraperitoneal injections. Furthermore, intraperitoneal injection is not an ideal drug delivery approach. Therefore, we plan to refine the experimental design and supplement the research results at the cellular level to enhance the universality and credibility of our conclusions.

## 4. Materials and Methods 

### 4.1. Animals

Healthy male wild-type C57BL/6J mice (3 weeks old, weighing 12–14 g) were obtained from Vital River Laboratory Animal Technology Co., Ltd., Beijing, China. The mice were allowed to acclimate to the new environment for at least three days before the commencement of the experiments. The mice were randomly assigned to four experimental groups (*n* = 10/group) and six blank control groups (*n* = 5/group). The experimental groups were as follows: FDM2M (FDM for 2 weeks with MCC950 injection), FDM4M (FDM for 4 weeks with MCC950 injection), FDM2W (FDM for 2 weeks with saline injection), and FDM4W (FDM for 4 weeks with saline injection). To prepare the MCC950 injectable solution, solid MCC950 powder (CP456773 and GLPBIO) was dissolved in freshly prepared saline solution at a concentration of 50 mg/mL. Injections were given on the first day of the experiment, followed by additional injections every other day. The final injection was given one or two days before the mice were euthanized. During the injection procedure, the mice were secured, and their abdominal area between the nape of the neck and the tail was fully exposed. The lower abdomen was punctured, and the injection solution was slowly administered to avoid a rapid increase in abdominal pressure. After the injection, the site was disinfected with alcohol to prevent infection. The volume of the MCC950 solution injected was 0.2 mL per kilogram of body weight. The control groups received an equivalent volume of saline solution. Among the six blank control groups, four corresponded to the experimental groups and underwent the same intraperitoneal injection procedure, but without the construction of the FDM model. The remaining two blank control groups did not undergo any procedures and were euthanized at the ages of 5 weeks and 7 weeks, respectively. The form-deprivation myopia (FDM) model was constructed by manually cutting non-toxic, semi-transparent PVC balloons to an appropriate size and positioning it over the right eye of each mouse. The edges of the balloon were then adhered to the mouse’s cheek, the midline area of the nose, and the fur below the ear using a non-toxic, quick-drying adhesive. This process ensured that the right eye was deprived of enough light stimulation, inducing a state of form deprivation. The balloon was adhered to the mouse’s eye at the start of the experiment. The mouse cage was observed twice daily to ensure that the balloon was promptly reattached in case of accidental detachment, ensuring that the eye exposure time did not exceed 12 h. The balloon remained in place until the end of the experiment, at which point the mice were euthanized. All mice were exposed to a 12 h light/dark cycle daily and provided with water, vegetables, and vitamins ad libitum. The mice were housed in a controlled environment with a relative humidity of 50%, a temperature of 22 °C, and a light intensity of 20 lux. Animal handling procedures adhered to the guidelines set forth by the Association for Research in Vision and Ophthalmology (ARVO) regarding the use of animals in ophthalmic and vision research. The animal experiments were conducted under pathogen-free conditions, and in accordance with the Institutional Animal Care and Use Committee (IACUC) protocol approved by Peking Union Medical College Hospital.

### 4.2. Assessment of Refractive Power and Axial Length

The mice were anesthetized with an intraperitoneal injection of 1% pentobarbital sodium. The experiment commenced three minutes after the mice became inactive, and was completed within a half-hour timeframe. To achieve rapid pupil dilation, a compound tropicamide solution was administered. Refractive power measurements of both eyes were conducted in a dark environment by aligning with the visual axis, using a streak retinoscope (Model YZ24, 66Vision Tech Co., Ltd., Suzhou, China). After euthanizing the mice, their ocular organs were carefully extracted using forceps, with an effort to meticulously remove surrounding connective tissues. The axial length (AL) was determined by aligning the eye along the eye axis on a screw micrometer. Measurements were recorded when the micrometer’s screw made contact with the corneal surface. This process was repeated three times, and the mean value was rounded to the nearest 0.001 mm for precision. All measurements were diligently conducted and recorded by experienced personnel.

### 4.3. Immunohistochemistry

The extracted eye samples were fixed in a 4% formaldehyde solution at 4 °C for 24 h, followed by sectioning, blocking, and incubation with primary antibodies. Subsequently, they were examined under a fluorescence microscope. For negative controls, normal serum was used in place of the primary antibodies. Evaluation of fluorescence intensities was carried out in a double-blind manner. The primary antibodies specifically targeting NLRP3 (1:200 dilution; #15101, Cell Signaling Technology, Danvers, MA, USA), caspase-1 (1:200; #24232, Cell Signaling Technology, Danvers, MA, USA), IL-1β (1:200; #12242, Cell Signaling Technology, Danvers, MA, USA), MMP-2 (1:200; ab86607, Abcam, Waltham, MA, USA), IL-18 (1:200; ab207323, Abcam, Waltham, MA, USA), and Collagen-1 (1:200; ab88147, Abcam, Waltham, MA, USA) were used in this study. The BOND Polymer Refined Detection (DS9800, Leica, Wetzlar, Germany) was employed for the staining procedure. Immunoreactivity scores were determined on a scale of 0 to 12, using a 13-point system. The score was obtained by multiplying the percentage of positive cells (PP) by the primary staining intensity (SI). PP values: 0 (no positive cells), 1 (<25% positive cells), 2 (25–50% positive cells), 3 (51–75% positive cells), and 4 (>75% positive cells). SI values: 0 (negative), 1 (weak), 2 (moderate), and 3 (strong).

### 4.4. Western Blotting

The sclera was snap-frozen in liquid nitrogen, ground into a fine powder, and then incubated in a lysis buffer for 1 h. The scleral lysate was prepared in an SDS lysis buffer containing a cocktail of protease and phosphatase inhibitors (Pierce, Rockford, IL, USA). Protein quantification was performed using the Bradford protein assay, and 40 μg of total protein samples was loaded onto sodium dodecyl sulfate-polyacrylamide gels at a concentration of 10%. After electrophoretic separation, the proteins were transferred onto a polyvinylidene fluoride membrane (Millipore, Billerica, MA, USA). The membrane was then incubated overnight at 4 °C with the same primary antibodies used in the immunohistochemistry experiments. The antibodies were all diluted at a concentration of 1:1500. For protein detection, horseradish peroxidase-conjugated secondary antibodies were used: goat anti-rabbit (ab6721, Abcam, Waltham, MA, USA) or anti-mouse (ab6789, Abcam, Waltham, MA, USA). The Western Blotting Reagents Kit (12957s, Cell Signaling Technology, Danvers, MA, USA) was employed for the detection process, followed by scanning with the Odyssey Fc system (LI-COR, Lincoln, NE, USA). Grayscale analysis was performed using ImageJ version 1.8 software. Relative grayscale values were obtained by normalizing the target band intensity to that of β-actin (1:1500; 66009-1-Ig, Proteintech, Tokyo, Japan). The experiment was conducted in triplicate, and the mean value was calculated.

### 4.5. Statistical Analysis

Statistical analyses were conducted with SPSS version 22.0 software. The normality of the measurement data was assessed using the Shapiro–Wilk test, and the data were expressed as mean ± standard deviation. Parametric tests, such as the t-test (for normally distributed data), Mann–Whitney U test (for non-parametric data), or Wilcoxon matched-pairs signed-rank test, were used for comparisons between two groups. For comparisons among three groups, a one-way analysis of variance was employed, followed by a post hoc test using Tukey’s test. A *p*-value of less than 0.05 was considered statistically significant.

## 5. Conclusions

In conclusion, the intraperitoneal injection of MCC950 can attenuate the progression of myopia in FDM by downregulating the expression levels of NLRP3 and its downstream signaling pathway factors (IL-1β, IL-18, Caspase-1, and MMP-2). This regulatory effect may involve MMP-2′s influence on Collagen-1. MCC950 holds promise as a potential therapeutic agent for controlling the progression of myopia.

## Figures and Tables

**Figure 1 ijms-24-15839-f001:**
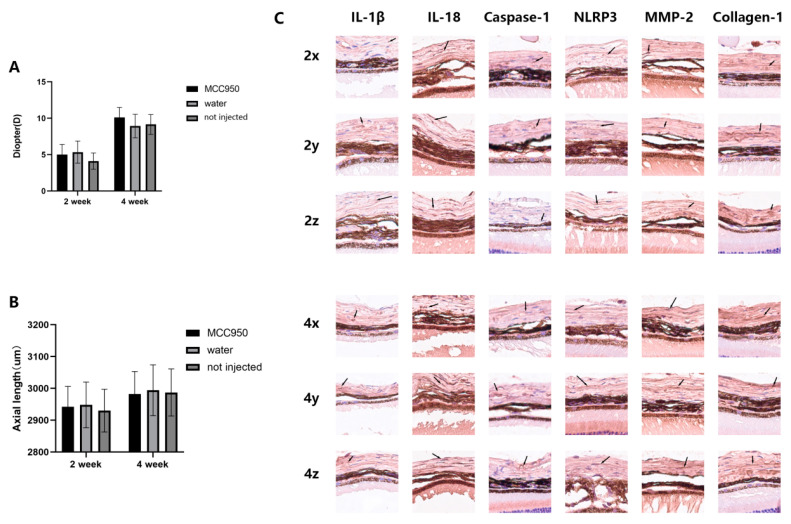
Refractive status and expression levels of NLRP3 signaling pathway-related factors in the three blank control groups of mice. The designation “2x” indicates that the eyes in this group are from the blank control group of the same age as the FDM2M group. Similarly, “2y” represents that the eyes in this group are from the blank control group of the same age as the FDM2W group, and “2z” represents that the eyes in this group are from the blank control group of the same age as the FDM2 group without intraperitoneal injection, and so on. No statistically significant differences were observed in the refractive status (**A**) and axial length (**B**) among the age-matched groups of mice. Immunohistochemical images of the mouse sclera (**C**) are presented, with each image corresponding to a size of 100 μm × 100 μm. Arrows indicate tissues stained by the respective antibodies.

**Figure 2 ijms-24-15839-f002:**
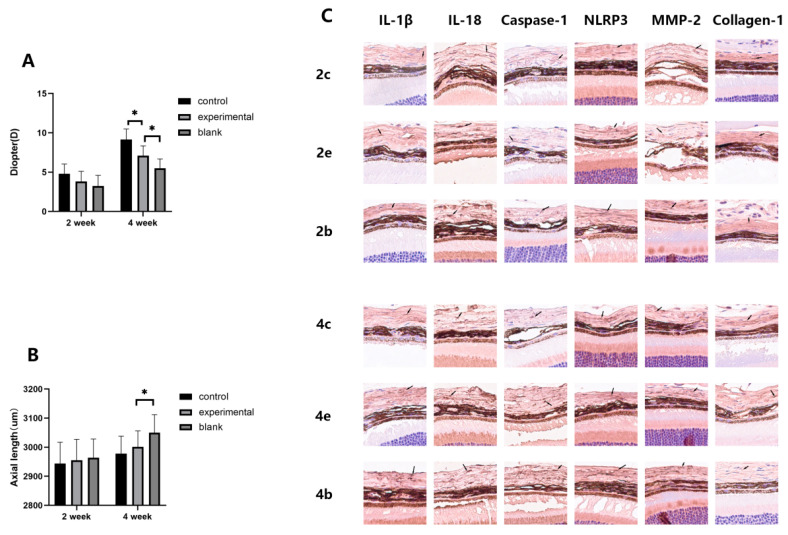
Refractive status and immunohistochemical expression of experimental eyes. The labels “2c”, “2b”, “2e”, and so on represent the origin of the mouse eyes within each group, with “2c” indicating control eyes from the FDM2M group, “2b” denoting experimental eyes from the FDM2W group, “2e” representing experimental eyes from the FDM2M group, and so forth. The results for refractive power are depicted in panel (**A**), while axial length measurements are presented in panel (**B**). Statistical differences between two groups are indicated by an asterisk (*) connecting the respective groups (i.e., Group A vs. Group B, *p* < 0.05). (**C**) Immunohistochemical images of the mouse sclera. with each image corresponding to an actual size of 100 μm × 100 μm. Tissues stained by the corresponding antibody are indicated by arrows.

**Figure 3 ijms-24-15839-f003:**
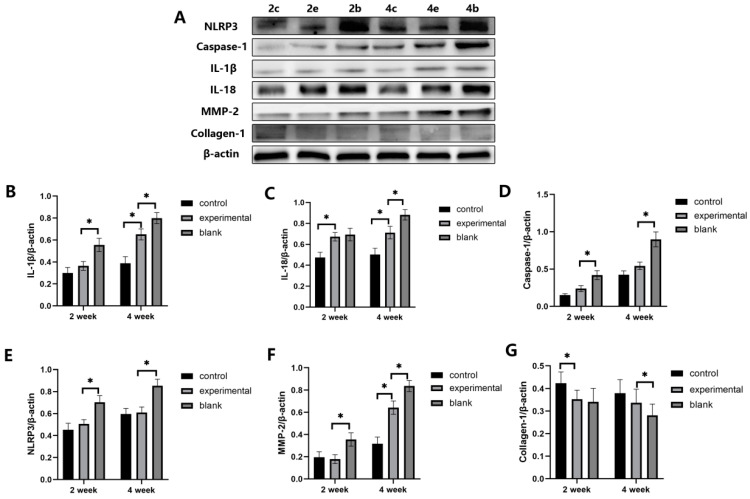
Injection of MCC950 affecting the expression levels of NLRP3 inflammasome-associated factors in the sclera of FDM mouse experimental eyes. The labels “2c”, “2b”, “2e”, and so on represent the origin of the mouse eyes within each group, with “2c” indicating control eyes from the FDM2M group, “2b” denoting experimental eyes from the FDM2W group, “2e” representing experimental eyes from the FDM2M group, “4e” representing experimental eyes from the FDM4M group, and so forth. (**A**) Western blot analysis reveals the expression levels of IL-1β, IL-18, caspase-1, NLRP3, MMP-2, and Collagen-1. (**B**–**G**) The semi-quantitative analysis results for IL-1β, IL-18, caspase-1, NLRP3, MMP-2, and Collagen-1, respectively. The data are presented as mean ± standard deviation (*n* = 3). Statistical differences between the two groups are indicated by an asterisk (*) connecting the respective groups (i.e., Group A vs. Group B, *p* < 0.05).

**Table 1 ijms-24-15839-t001:** Immunoreactivity scores of the sclera of wild-type mice in the blank control group (*n* = 5).

	2x	2y	2z	4x	4y	4z
IL-1β	1.9 ± 1.3	2.4 ± 1.0	1.4 ± 1.2	2.9 ± 1.8	3.2 ± 2.0	2.3 ± 1.4
IL-18	2.1 ± 1.6	2.8 ± 1.3	2.5 ± 1.2	4.9 ± 2.1	5.5 ± 1.4	5.6 ± 2.9
Caspase-1	2.8 ± 1.5	3.3 ± 1.4	2.0 ± 0.7	3.7 ± 1.8	4.1 ± 1.7	3.0 ± 1.6
NLRP-3	2.4 ± 0.9	3.6 ± 1.1	2.8 ± 0.9	2.9 ± 0.9	4.6 ± 1.3	4.2 ± 1.0
MMP-2	2.4 ± 1.2	2.9 ± 1.5	1.7 ± 1.0	3.1 ± 1.3	4.9 ± 1.5	4.5 ± 1.6
Collagen-1	5.0 ± 2.0	4.7 ± 1.4	4.8 ± 1.7	4.8 ± 1.5	4.3 ± 1.1	4.2 ± 1.7

**Table 2 ijms-24-15839-t002:** Immunoreactivity scores of the sclera of wild-type mice with or without MCC950 injection (*n* = 5).

	2c	2e	2b	4c	4e	4b
IL-1β	1.7 ± 0.8	2.4 ± 1.0	4.6 ± 1.2	2.8 ± 1.2	6.8 ± 2.0	8.4 ± 2.6
IL-18	4.1 ± 1.2	5.7 ± 1.3	6.5 ± 1.5	3.8 ± 1.1	5.5 ± 1.4	8.1 ± 2.4
Caspase-1	2.8 ± 1.0	3.1 ± 1.2	3.9 ± 1.4	4.0 ± 1.1	6.5 ± 1.7	8.0 ± 1.8
NLRP-3	4.0 ± 1.4	5.5 ± 1.1	6.9 ± 1.6	4.8 ± 1.3	5.4 ± 1.3	7.8 ± 2.0
MMP-2	1.9 ± 0.9	2.2 ± 1.0	3.4 ± 1.2	3.7 ± 1.3	6.3 ± 1.5	7.7 ± 1.9
Collagen-1	5.3 ± 1.4	5.0 ± 1.4	3.7 ± 1.2	4.8 ± 0.8	3.9 ± 1.1	2.7 ± 1.0

## Data Availability

Not applicable.

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
