# Peer review of "Intraperitoneal Injection of MCC950 Inhibits the Progression of Myopia in Form-Deprivation Myopic Mice"

_ijms, 2023, doi:10.3390/ijms242115839_

Round 1

Reviewer 1 Report

Comments and Suggestions for Authors

Summary:

NLRP3 is an inflammatory mediator shown to increase MMP-2 expression in the sclera, thus leading to decreased collagen-1 in the sclera with subsequent scleral remodeling and possible development of axial myopia. MCC950 inhibits NLRP3 and thus may be a therapeutic pathway in preventing or treating myopia. Form-deprivation was performed by covering the right eyes of a set of mice with a PVC balloon adhered to the fur. Intraperitoneal injections of MCC950 or saline were administered to groups of mice over two or four weeks to FDM and control groups. Animals were anesthetized for assessment of refractive power via a streak retinoscope, then euthanized. Axial length was measured using a screw micrometer. Eye tissues were fixed in formaldehyde for immunofluorescent analysis of NLRP3, caspase-1, MMP-2, IL-1B, IL-18, and collagen-1 expression. These targets were also measured for protein quantification via Western blot analysis of scleral tissue. Overall, the authors found that MCC950 injection in the FDM mice led to less severe axial length elongation, higher expression of collagen-1, and lower expression of NLRP3, MMP-2, caspase-1, IL-1B, and IL-18 compared to saline-treated FDM mice. Small changes in expression were found between control mice and the MCC950-treated mice.

Review:

The authors did a commendable job in thoroughly investigating the major proteins in this inflammatory pathway. The methodology included ample experimental and control groups, and the data was measured with multiple modalities for completeness. The results add to our understanding of the pathogenesis of myopia. The conclusions drawn by the paper are appropriate and well-contextualized with the current literature with discussion of current medications and their effectiveness. At this time, it is uncertain if MCC950 will be a realistic clinical intervention, but further understanding of the biochemical mechanisms of disease is essential for medical progress.

For the methods section, it is unclear when the FDM model was constructed relative to the rest of the experiment. I believe that the PVC balloons were adhered to the mice on day 1 of the experiment, but this should be clarified in the text.

For the FDM model, the right eye was obstructed to induce myopia. Was any analysis done on the left eyes? It would be interesting to see if any changes in scleral structure or protein expression were found in the left eyes of the FDM model or if these left eyes were identical to the eyes from the relevant control groups. Even without this additional data, the paper overall has good scientific merit and is worthy of publication with minor edits.

Comments on the Quality of English Language

Further editing is needed to fix incomplete or unclear sentences, particularly in the introduction:

·         Lines 20-21: This sentence has redundant words (the phrase “were divided into experimental groups” on line 20 should be removed).

·         Line 69: The series of arrows is misleading. Arrows typically imply a chain of positive cell-signaling/activation, whereas the sentence is explaining a series leading to decreased collagen-1. Adding vertical up- or down-arrows adjacent to the proteins would help clarify the activity in the pathway.

·         Line 75: The comma is grammatically unnecessary.

·         Line 79: “Importantly, it does not with essential…” This sentence needs another verb to clarify what the compound does not do.

·         Lines 80-83: These sentences are incomplete and run together.

·         Line 96: “Mcc950” should be written as “MCC950” for consistency within the paper.

There is inconsistent capitalization of “Collagen-1” versus “collagen-1” throughout the paper. Also, there is inconsistent hyphenation of “form-deprivation” versus “form deprivation” throughout the paper.

Author Response

Thank you very much for your valuable feedback! Regarding the issues you raised: 
1. The balloon was indeed adhered to the mouse at the beginning of the experiment, and I have provided additional details in the "Methods" section of the main text (The balloon was adhered to the mouse's eye at the start of the experiment. The mouse cage was observed twice daily to ensure that the balloon was promptly reattached in case of accidental detachment, ensuring that the eye exposure time did not exceed 12 hours).
2. I apologize for not fully understanding your question. If you are referring to the left eye of the mice injected with MCC950, then the images in Figure 2c and 4c of this study are what you are looking for. However, if you are referring to the left eye of the normal FDM mice without MCC950 injection, I have previously measured this in another experiment, but due to copyright reasons, I cannot include the images in this paper. If you are interested, you can refer to reference 14, which provides specific details. In summary, the expression levels of various factors and the degree of myopia in the contralateral eye of FDM mice showed no significant differences compared to the control group.
3.I have made the necessary revisions to address the English expression issues you raised and have submitted it to my English editor as per your request. However, it will take some time for the modifications to be made, so this response does not include any content polishing. I will perform a more thorough editing before the final version is finalized. 
Thank you once again for your valuable suggestions.

Reviewer 2 Report

Comments and Suggestions for Authors

The study by Chen et al. describes the use of the inhibitor MCC950 to reduce the disease phenotype associated with axial myopia.  The data is well presented although the use of "2b, 2c" makes the figure legends difficult to understand. Overall the study demonstrates a significant effect, but demonstrates little more, and is a bit lightweight in the conclusions.

1) In the abstract "divided into experimental" on line 20 is repeated twice in the sentence.

2) Line 79 "Importantly, it does not" doesn't make much sense and should be re-written

3) In the Figure 3 legend please define 4e and 4b

4) The manuscript would be easier to understand if there was a graphic illustration of the signaling process that is targeted.

Comments on the Quality of English Language

With a few minor exceptions, the manuscript is well written.

Author Response

3. Regarding the ‘lightweight’  and lack of graphic illustration, there are the following reasons: This paper is the third part of my doctoral research project. The first two parts aimed to demonstrate the correlation between high expression of NLRP3 and myopia in mice, as well as the causative role of NLRP3 in myopia development. Originally, these three parts were intended to be merged into a complete article. However, due to unfinished work on MCC950 before graduation, I had to publish the first two parts separately for the purpose of completing my degree. As a result, the third part, presented separately, cannot use the mechanism diagram from the previous sections and overall content appears relatively limited. In future research, I will explore other possible administration methods for MCC950 to further investigate its control effects on myopia in mice, with potential clinical translation in mind.